# Significance of Conserved Regions in Coronavirus Spike Protein for Developing a Novel Vaccine against SARS-CoV-2 Infection

**DOI:** 10.3390/vaccines11030545

**Published:** 2023-02-24

**Authors:** Titus A. Olukitibi, Zhujun Ao, Bryce Warner, Rodrigo Unat, Darwyn Kobasa, Xiaojian Yao

**Affiliations:** 1Laboratory of Molecular Human Retrovirology, Winnipeg, MB R3E 0J9, Canada; 2Department of Medical Microbiology, Max Rady College of Medicine, Rady Faculty of Health Sciences, University of Manitoba, Winnipeg, MB R3E 0J9, Canada; 3Special Pathogens Program, National Microbiology Laboratory, Public Health Agency of Canada, Winnipeg, MB R3E 3R2, Canada

**Keywords:** SARS-CoV-2, vaccine, spike protein, mutation, conserved epitopes

## Abstract

Over the years, several distinct pathogenic coronaviruses have emerged, including the pandemic SARS-CoV-2, which is difficult to curtail despite the availability of licensed vaccines. The difficulty in managing SARS-CoV-2 is linked to changes in the variants’ proteins, especially in the spike protein (SP) used for viral entry. These mutations, especially in the SP, enable the virus to evade immune responses induced by natural infection or vaccination. However, some parts of the SP in the S1 subunit and the S2 subunit are considered conserved among coronaviruses. In this review, we will discuss the epitopes in the SARS-CoV-2 S1 and S2 subunit proteins that have been demonstrated by various studies to be conserved among coronaviruses and may be immunogenic for the development of a vaccine. Considering the higher conservancy of the S2, we will further discuss the likely challenges that could limit the S2 subunit from inducing robust immune responses and the promising approaches to increase its immunogenicity.

## 1. Introduction

In the history of humans, the Coronavirus disease (COVID-19) caused by the severe acute respiratory syndrome coronavirus-2 (SARS-CoV-2) has made an indelible mark characterized by a contagious respiratory pandemic [1,2]. Consequently, 660 million cases and approximately 6.6 million deaths have been reported as of December 2022 [3]. Since its emergence, studies have been focusing on developing preventive and therapeutic measures against this disease, and several vaccines have been licensed [4,5]. However, the constant mutation of the SARS-CoV-2 virus, which results in different variants, narrows the effectiveness of the licensed vaccines against SARS-CoV-2 infections, thus suggesting a needed continuous effort in developing a SARS-CoV-2 universal vaccine [6]. Besides the pandemic caused by SARS-CoV-2, other human coronaviruses (huCoVs) including human CoV (HCoV)-229E (1962), HCoV-OC43 (1967), SARS-CoV (2002), HCoV-NL63 (2004), HCoV-HKUI (2005), Middle East respiratory syndrome coronavirus (MERS)-CoV (2012) and SARS-CoV-2 (2019) have been implicated in different outbreaks since the start of the 21st century [7,8,9,10,11]. Although the trend of the emergence of coronaviruses remains unclear, adequate preparation must be made for a possible emerging or re-emerging strain. In this case, developing an effective universal vaccine against CoVs should take advantage of the conserved portions of the virus, especially the spike protein (SP).

Levels of similarities have been observed among the emerged CoVs [12,13,14]. For instance, the SARS-CoV-2 viral genome sequence analysis revealed its phylogenetic similarity with SARS-CoV (79%) and MERS-CoV (50%) [15,16,17]. Like other betacoronaviruses, the SARS-CoV-2 genome consists of 27 proteins encoded by 14 open reading frames (ORF), including non-structural proteins (nsp) that are encoded by the ORF 1 and 2 present at the 5′-terminal regions [18,19]. More importantly, the SARS-CoV-2 structural proteins, notably the SP, envelope protein (E), membrane protein (M), nucleocapsid (N) and eight accessory proteins encoded by the 3′-terminal region of the genomes, are in like manner with other betacoronaviruses [19].

Sequence analysis of the SARS-CoV-2 SP revealed a close association with SARS-CoV regarding amino acid composition as well as comparable binding affinity to human angiotensin-converting enzyme 2 (hACE2) [20,21]. The SP, which is a highly N-glycosylated class I transmembrane fusion protein, plays a critical role during coronavirus infection. The SP mediates the attachment of the virus into the cell receptor and facilitates the viral-host membrane fusion [22,23]. The SP also assembles into trimers on the virus surface and cleaves into two subunits, S1 and S2, during the cell infection [24,25]. The large protein S1 domain contains the RBD that is responsible for binding to the host cell receptor and varies extensively in all isolates of the CoVs [20,24,25]. Meanwhile, the S2 domain, which facilitates the virus-cell fusion, is made up of the fusion peptide (FP) and heptad repeat regions (HR1 and HR2), which are conserved among the isolates of the CoVs [26,27], the conserved membrane-proximal external region (MPER) and the transmembrane domain (TM) [28,29,30,31,32,33,34]. After the binding of the S1 RBD domain to the ACE2, the S2 subunit then inserts its FP into the cell membrane leading to the assemblage of the HR1 and HR2 into a six-helix structure to drive the cellular and viral membrane closely together for viral entry (Figure 1) [35]. Notably, the major determinant of cell tropism in most coronaviruses is typically linked to the structure of the SP [21,25,36,37,38,39,40]. Phylogenetic, bioinformatic and homology structural modelling analyses showed that the RBD of SARS-CoV-2 only has 64% shared identity with the SARS-CoV, while the NTD has 51% similarity [41]. However, the study revealed that within the S2, the fusion protein (FP) is 93% identical and the HR1 is 88% identical, while the HR2 and the TM are respectively 100% and 93% similar.

SARS-CoV-2 and other SARS-related coronaviruses (e.g., SARS-CoV) can utilize distinct domains within the S1 subunit to identify different attachment and entry receptors in the host cell surface [20,40]. For instance, the differences in the SPs of the known huCoVs contribute to their difference in pathogenesis and the site of infection (lower or upper respiratory) [36]. Taking a closer look at SARS-CoV-2 as an example, Laporte et al. described that the higher transmissibility experienced with the SARS-CoV-2 compared to other human coronaviruses is due to its abundant replication in the upper respiratory tracts [43]. They mentioned that the SARS-CoV-2 SP has an intrinsic temperature preference of 33 °C, like the temperature of human respiratory tracts, instead of the 37 °C required by other human CoVs. In addition to this, it was revealed that the SARS-CoV-2 has multiple cell entry activators, including TMPRSS2 and TMPRSS13 protease, broadening its tropism. As mentioned by Korber et al., a D614G mutation observed in the SARS-CoV-2 variant S1 also resulted in the wider spread of the virus at different geographical regions with higher viral loads in the respiratory tracts [44]. These differences observed in the SP of the CoVs contribute to the difficulties in developing a sustainable vaccine against the emerging variants of SARS-CoV-2. It is therefore a necessity to develop a vaccine that can prevent the spread of all present or emerging variants of SARS-CoV-2.

Notably, certain conserved epitopes in the S1 and S2 regions can be utilized to develop universal vaccines against CoV infections [45,46,47]. However, some varying challenges surround these conserved epitopes, including the presence of immunodominant epitopes [45], weak immunogens and non-neutralizing antibodies [47]. This review aims to shed light on the conserved epitopes capable of inducing a broad immune response against multiple CoVs and highlight how to circumvent the associated possible challenges with lessons gleaned from SARS-CoV-2.

## 2. Impact of Mutations on the SARS-CoV-2 SP to the Evasion and Resistance of Immune Responses

Understanding the possible ways by which the SARS-CoV-2 SP escapes immune responses will shed light on the importance of using the conserved regions of the SP in developing a universal vaccine against variants of coronaviruses. The SARS-CoV-2 SP has been demonstrated to regulate the innate immune response through the activation of the NF-κB pathway in human macrophages and monocytes, thus increasing the production of inflammatory responses (IL-8) [22]. Continuous mutation of the SARS-CoV-2 SP impacts innate immune responses and could further increase virulence, pathogenicity and immune evasion of the virus given that one of the strategies by which SARS-CoV-2 evades immune response is through cytokine shock [48,49]. Noteworthily, mutations in the SARS-CoV-2 ORF8 (which could modulate the SP during transmission) have been associated with inhibition of the Major Histocompatibility Complex (MHC)-1 stimulation, thus leading to the low production of cytotoxic T cells (CD8^+^ T cells) [50,51,52]. In our study, the SARS-CoV-2 Delta variant SP had the highest potential of stimulating the NF-κB pathway in macrophages to induce inflammatory cytokines in the monocyte-derived macrophages (MDMs) and monocyte-derived dendritic cells (MDDCs) when compared with the Wuhan strains or D614G variants. This suggests that continuous mutation of the SARS-CoV-2 SP also impacts innate immune responses and could further increase virulence, pathogenicity and immune evasion of the virus [48].

The adaptive immune response regarded as the second line of defence during infection is regulated by the antigen-presenting cells (APCs) and comprises the antibodies (B cells), T helper cells (CD4^+^ T cell) and CD8^+^ T cells [53]. In the case of SARS-CoV-2 infection, the DCs present peptides derived from the SP on the MHC-I or II for the activation of the specific T-cells (CD4^+^ and CD8^+^) [54]. The SARS-CoV-2 specific CD4^+^ T and CD8^+^ T cells help to trigger the induction of SARS-CoV-2 specific B cells that further generate most of the neutralizing antibodies as well as memory B cells and long-term humoral immunity. They also regulate antiviral activities, recruit innate cells, kill infected cells, and facilitate tissue repair [53,55,56,57,58,59]. Notably, among the SARS-CoV-2 structural proteins, the SP is the most promising antigen for the induction of neutralizing antibodies (including IgM, IgA and IgG) and cell-mediated immune responses (including CD4^+^ T, CD8^+^ T cells and T follicular helper cells (Tfh)) during COVID-19 disease [60,61,62,63,64,65,66,67,68].

However, the SARS-CoV-2 SP could escape humoral immune responses and neutralizing antibodies because of mutations in its genetic constituent, thus leading to the ineffectiveness of vaccine-induced immunity [21,69]. Vaccine development against SARS-CoV-2, like all other vaccines, depends on the induction of neutralizing, adaptive, and memory immune responses. Although the current COVID-19 vaccine reduces the severity of the disease, the re-infection of SARS-CoV-2 in previously infected people or fully vaccinated individuals has been well-documented, suggesting that the sterilizing immunity towards SARS-CoV-2 is partial, short-lived and narrow [70,71,72].

Since the emergence of the SARS-CoV-2 virus in 2019, various variants have emerged with mutations leading to amino acid changes in the SP which contribute largely to immune evasion [73,74]. Most of the mutations are found in the S1 subunit protein. SARS-CoV-2 variants include the: Alpha, having the N501Y mutation in RBD; Beta, having N501Y, K417N and E484K mutations in the RBD; Gamma, with N501Y, K417T and E484K; Delta, having T478K and L452 in the RBD and Omicron with S371L, G339D, S375F, S373P, K417N, N440K, S477N, G446S, E484A, T478K, Q493K, Q498R, G496S, N501Y and Y505H in the RBD (Figure 2) [74,75]. Among all, Omicron variants and subvariants contain the highest number of changes in the SP relative to the ancestral virus. Compared to the wild type, Delta has 9 mutations, including 3 RBD mutations (K417N, E484Q and L452R) and D614G in the SP, whereas Omicron has more than 32 mutations in the SP with 15 mutations in the RBD, thus influencing its transmissibility and infectivity [76]. The T95I, N211I, V213R, Y505H, N786K and N856K mutations in Omicron variants contribute to its pathogenicity and virulence [77]. Sub-variants of Omicron have also been identified as BA.1, BA.2, BA.2.12.1, BA.2.3, BA.2.9, BA.3, XBB, BA.4, BA.5 (BQ.1, BQ.1.1, BF.7 and BA.4.6) and recombinant BA.1/BA.2 [78,79]. The BA.2 progressively replaced BA.1, which was the most prevalent globally, while BA.3 spread is restricted. The BA.4 and BA.5, which were first isolated in South Africa, replaced the BA.2 variant, having mutations F486V and R493Q and deletion in positions 69 and 70 in the RBD [80,81].

The mutations on the RBD impact its ability to bind with the host ACE2 receptors. For instance, the RBD of the SARS-CoV-2 has a greater affinity for the hACE2 than the RBD of SARS-CoV due to the differences in their amino acids [24]. These following mutations have been identified as the RBD mutations of concern present in the VOCs resulting in the compromising of immune responses by the Food and Drug Agency (FDA): P337H/L/R/T, K417E/N, E484K/Q/P/D, N439K, K444Q, S494P, V445A, N450D, L452R, Y453F, E340A/K/G, L455F, F486V, N460K/S/T, D420N, V483A, F490S, Q493K/R and N501Y/T [82]. The Omicron subvariants shared many mutations among themselves including the P681H, H655Y, N679K and D614G mutations in the S1 subunit; G339D, S371L/F, S373P, S375F, K417N, N440K, S477N, T478K, E484A, Q498R, N501Y and Y505H in RBD; G142D in N-terminal domain (NTD) and N969K, N764K, D796Y and Q954H in the S2 subunit (Figure 2) [83]. Omicron BA.1 and BA.2 have 12 common mutations on the RBD, which are G339D, S373P, S375F, K417N, N440K, S477N, T478K, E484A, Q498R, Q493R, N501Y and Y505H, whereas BA.2 has a total of 31 mutations in the SP including the unique R408S, and S371F (Figure 2). However, the BA.3 also has unique mutations T376A and D405N in the RBD [81].

A study by Hoffmann et al. showed that the Delta variant with E484Q mutation on the RBD had a reduced neutralizing sensitivity to bamlanivimab or plasma from vaccinated patients [84]. Moreover, the N439K has been demonstrated by Thomson et al. to enhance infectivity and escape humoral immune response by enhancing the binding of the RBD to hACE2 [85]. Interestingly, before the emergence of the Omicron variant, a CR3022-neutralizing antibody isolated from a SARS-CoV-2 convalescent patient was revealed to target highly conserved epitopes of the RBD of SARS-CoV-2 and SARS-CoV [73]. However, with the emergence of the Omicron variants which contain 15 mutations on the RBD, the sensitivity of CR3022 was reduced [86]. Aside from the impacts of the mutations on the RBD in the escape of the immune response, the D614G mutation on the non-RBD of the S1, found in most VOCs, also contributes to the enhancement of the virulence and immune evasion of SARS-CoV-2 variants [22,44].

Therefore, it is not surprising to find that other variants of SARS-CoV-2, such as Beta (B.1.351), Gamma (B.1.1.28), Delta (B.1.617.2) and especially Omicron (B.1.1.529), substantially escape the immune responses induced by the licensed vaccines [22,86,87,88]. Supportably, a report showed that 21 out of 33 people that have been vaccinated 3 times with the licensed vaccine were still susceptible to the Omicron variants’ infection [89]. By using a pseudotyped lentivirus system [90], a study also revealed that the Omicron SP had 26-fold resistance to neutralizing antibodies among the convalescent donor and 26- to 34-fold resistance to antibodies elicited by the Pfizer BNT162b2 and Moderna two dosage vaccines. In addition, the Omicron spike resisted most therapeutic antibodies except sotrovimab and evaded immune responses induced in the individual vaccinated with BNT162b2, with 12–44-fold resistance compared to the Delta variant. Similarly, the third dosage of BNT162b2 or vaccination with ChAdOx1 induced neutralizing antibodies against the Omicron, but fewer than against the Delta [91]. The observed compromise of the immune response by the variants is particularly due to the mutations on the RBD of the S1 [89,92].

Importantly, the Omicron BA.1, BA.2 and BA.3 are very good at escaping immune responses induced by natural infection or by one or two doses of the vaccine, while a third dose can give some protection [79,93,94]. Although the SARS-CoV-2 bivalent vaccine (wildtype and Omicron) can provide long-term protection, a BA.1 derived vaccine may not provide broad protection against SARS-CoV-2 variants [95]. Furthermore, the Omicron BA.4, BA.5 and BA.2.12.1 have been demonstrated to show more immune-response escape strategies than the BA.1 and BA.2 due to the presence of the F486V and D405N mutations [86,95]. Cao et al. [95] demonstrated that, structurally, the SP of BA.4, BA.5 and BA.2.12.1 have the same binding affinity with the ACE-2. They further revealed that BA.4, BA.5 and BA.2.12.1 could evade neutralizing antibodies from the plasma of BA.1 infected immunized patients or triple-vaccinated individuals to a greater extent than BA.2.

The SARS-CoV-2 S2 subunit protein also has about 12 mutations which are not shared among the variants and may influence the immune evasion of SARS-CoV-2 [96]. Nevertheless, the S2 region is less affected by the spontaneous mutation because of its importance in the fusion process that leads to infection [97]. Indeed, despite the variation impacted by mutations of the SP of the CoVs, some certain epitopes that could induce neutralizing antibodies, memory B cells and memory T cell responses remain conserved in the S1 and S2 subunits across the CoVs. Therefore, attention should be placed on these conserved regions for developing a universal vaccine.

## 3. SARS-CoV-2 S Conserved Regions as a Potential Target for Vaccine Development

Developing a vaccine for viruses with multiple strains or variants tends to take advantage of the conserved epitopes present in the virus. Conserved epitopes are epitopes that are relatively the same among different strains of a pathogen. For example, due to the multiple strains of influenza over the years, the means of developing a universal vaccine has been the use of the conserved epitopes on the hemagglutinin (HA stalk) or the matrix ectodomain (M2e) [98,99,100,101,102,103,104]. The strategy of using conserved epitopes has also been used in the development of preventative vaccines for HIV, dengue virus, Lassa fever virus (LASV), hepatitis virus and Kaposi’s sarcoma-associated herpesvirus (KSHV) [105,106,107,108,109,110,111,112,113]. Therefore, identifying the conserved regions in the SP of SARS-CoV variants will be of great importance in developing a universal vaccine against CoVs. Although mutations in SARS-CoV-2 had a great impact on the SP, studies have revealed that certain conserved epitopes in S1 can induce neutralizing antibodies [114]. Interestingly, some monoclonal antibodies (including 7B11, 18F3, S309 and its Fab, S315, 154C, S304, 240C and VHH-72) that recognize SARS-CoV and MERS-CoV could cross-react and cross-neutralize SARS-CoV-2 by recognizing the ACE2 binding sites on the SARS-CoV-2 RBD [115].

Jaiswal et al. mathematically (in-house developed PERL scripts) revealed sets of epitopes on the S1 subunit around 453 to 538 that interact with the ACE and are 99% conserved among the variants of SARS-CoV-2 [116]. Their study further identified conserved common neutralizing epitopes on the SARS-CoV-2, including YLTPGDSSSGWTAGAAAYYV (247–267 aa), TFKCYGVSPTKLNDL (376–390 aa) on S1 and LNEVAKNLNESLIDLQELGK (1186–1205 aa) on the S2 [116]. A recent immunoinformatic study predicted conserved and highly immunogenic CTL-induced epitopes on S1 VRFPNITNL (327–335 aa) and PYRVVVLSF (507–515 aa) (Table 1), while the CTL-induced epitopes on S2 were identified to be VVFLHVTYV (1060–1068 aa) and GVVFLHVTY (1059–1067 aa) (Table 2) [117]. Based on the conservancy, antigenicity, allergenicity, population coverage and transmembrane location, another study chose potential conserved epitopes from the S1 (FNATRFASVYAWNRK, 342–356 aa) (Table 1), S2 (FLHVTYVPAQEKNFT, 1062–1072 aa) (Table 2) and the E/M protein to construct a SARS-CoV-2 vaccine and revealed that it had high immunogenicity and broad neutralizing activity against the SARS-CoV-2 RBD (Figure 3) [118]. Jiang et al., with web-based analytic tools, also predicted potential T cell epitopes induced by the SARS-CoV-2 SP and narrowed them down to CD4 or CD8 T cell epitopes using ELIspot and a cytolytic assay [119]. In their observation, YYVGYLQPRTFLLKY (264–278 aa), located at the end of the NTD and the upstream of the RBD, is highly conserved among 11 variants of SARS-CoV-2 VOCs and variants of interest (VOIs) and could induce the T cells. Further studies also suggested that these epitopes could be well recognized by most HLA alleles globally [119].

Considering the role played by the S2 subunit during SARS-CoV-2 infection, it could also be targeted to induce immune responses against SARS-CoV-2. Interestingly, Ladner et al. generated an epitope-resolved analysis of IgG cross-reactivity among all CoVs in COVID-19-negative and recovered patients using a highly multiplexed peptide assay (PepSeq) and discovered that the epitopes at the FP, which is 93% similar among strains of betacoronaviruses and alphacoronaviruses, produced broadly neutralizing antibodies against the endemic coronaviruses, including SARS-CoV, MERS-CoV and SARS-CoV-2 [41,125]. Likewise, the HR2 region of the S2 is 100% conserved among the variants of the SARS-CoV-2 [41,128]. The S2 subunit proteins could also induce cross-reactive antibodies against the SARS-CoV SP and the endemic CoVs [130]. The S2 has been reported to induce neutralizing antibodies or T cell responses targeting the FP proximal region and HR2 domain of S2 in COVID-19 patients or animals vaccinated or infected with different CoVs [131,132,133,134]. Interestingly, due to prior population exposure to common cold coronaviruses, nAbs and memory B and T cells against SARS-CoV-2 were found in some individuals who have never been infected by SARS-CoV-2 [135,136]. Pre-existing antibodies against conserved epitopes of S2, such as residues 901–906, 810–816, 851–856, 1040–1044 and 1205–1212, showed the greatest cross-reactivity and hindered SARS-CoV-2 entry into cells [135]. Other identified regions that are most widely recognized among SARS-CoV-2 linear epitopes in convalescent donors are EELDKYF (1150–1156 aa) within the stem helix of the HR2 terminal and EDLLFN (819–824 aa), which overlaps the FP and is adjacent to the S2 cleavage [125,126]. Moreover, a study conducted by Song et al. revealed that monoclonal antibody CC9.3 isolated from individuals before SARS-CoV-2 infection was characterized to recognize the S2 subunit of the SARS-CoV-2 and other huCoVs [137].

From the sera of patients recovered from SARS-CoV-2, Pinto et al. isolated five monoclonal antibodies that could recognize the stem-helix (SH) of the S2 subunits of other betacoronaviruses including the OC43 strain [126]. In addition, Lu et al. identified and crystallized T cell follicle helper cells (cTfh) among patients that recovered from the mild symptoms of COVID-19 and revealed that these cTfh could recognize SARS-CoV-2 S2 subunit epitopes (864–882 aa) that are conserved among the emerging variants [57]. In a recent publication, Wu et al. identified a monoclonal antibody hMab5.17 that could recognize the SARS-CoV-2 HR2 domain that is adjacent to TM (SPDVDLGDISGINAS; 1161–1175 aa) and could protect against SARS-CoV-2 in the Syrian hamster. They further cloned the mAb hMab5.17 and demonstrated that it could neutralize SARS-CoV-2 variants [128].

Another highly conserved region located at S2 region is the SARS-CoV-2 MPER-like region (MPER) (GKYEQYIK; 1204–1211 aa), which has great potential for being used as an antigen for broadly neutralizing antibodies (bnAbs) [34,76,138]. Yu et al. have demonstrated that a cholesterol-conjugated lipopeptide containing SARS-CoV-2 MPER prepared by using cholesteryl succinate monoester could inhibit viral entry, indicating the importance of MPER in viral entry and fusion [34]. Like the MPER of HIV-1, the MPER of SARS-CoV-2 could possibly induce bnAbs against the variants of SARS-CoV-2. Studies have shown that bnAbs 4E10, 2F5, 10E8 and LN01 could interact with the MPER of the HIV-1 gp41 to prevent infection [139,140,141]. Likewise, the SARS-CoV-2 MPER could be a suitable immunogen for inducing neutralizing antibodies [34,138].

The above findings suggested that the S2 subunit of the SARS-CoV-2 is conserved among the previous strains of human coronaviruses and the SARS-CoV-2 variants and can induce cross-reactive antibodies.

## 4. Possible Challenges and Promising Approaches with the Conserved SARS-CoV-2 S2 Subunit in Vaccine Development

Structural positioning and immunodominance: The structural positioning of the S2 subunit might limit its ability to induce sterilizing immunity. The S2 is hidden under the S1 subunit protein, thereby being masked by the S1 subunit protein, resulting in the induction of weak immune responses during natural infection or when the whole spike protein is used in the vaccine development [142]. Studies have demonstrated that a vaccine targeting the S2 can induce IgG, but in a lesser amount when compared with the S1 subunit protein [143,144]. Wang et al. also demonstrated that the S1 of MERS-CoV in a DNA-based vaccine regimen elicited more neutralizing antibodies than the S2 subunit of the MERS-CoV vaccine [145,146]. Notably, the S1 could induce stronger neutralizing antibodies, but they might not be as broad due to mutation; however, the S2 neutralizing epitopes can induce broader neutralizing antibodies, but they may not be as strong as the S1 epitopes. Using an in vitro pseudotyped neutralization assay, a study revealed that combined human monoclonal antibodies against the HR1, HR2 and S1 of SARS-CoV had better cell-entry inhibition compared with the human monoclonal antibody against HR2. Interestingly, human monoclonal antibodies against HR1 or HR2 of the SARS-CoV have more broadly neutralization activity against different strains of human coronaviruses than the monoclonal antibodies of the S1 ectodomain [147], suggesting the SARS-CoV S2 subunit protein is a very promising epitope for developing a universal vaccine against the various strains of coronaviruses.

The immunodominance epitopes on the S1 subunit protein could also contribute to the induction of immune responses toward the S2. The analysis of the immunodominant and immunoprevalent SARS-CoV-2 epitopes of CD4 T cell or CD8 T cell revealed that SARS-CoV-2 has conserved immunodominant epitopes in the SP, M and the ORF1 [148,149,150]. Three immunodominant epitopes (TRFASVYAWNRKRISNCVAD; 345–364 aa, DEVRQIAPGQTGKIADYNYK; 420–439 aa and ERDISTEIYQAGSTPCNGVE; 480–499 aa) located at the SARS-CoV-2 RBD have been identified [151]. The epitopes ^350^VYAWN^354^ and ^407^VRQIAP^412^ are highly conserved for the variants of SARS-CoV-2 and SARS-CoV, while ^473^YQAGSTP^479^ within the RBD is only conserved among strains of SARS-CoV. In addition, Polyiam et al. also highlighted some immunodominant epitopes in the SARS-CoV-2 RBD, including NNLDSKVGGNYNYLYRLFRKSNLKPFERDISTEIYQAGST (439–478 aa.) and LFRKSNLKPFERDISTEIYQAGST (455–478 aa.) [152].

Short epitopes and low immunogenicity: Another possible challenge for using the S2 subunit protein as a universal vaccine is the shortness of the highly conserved epitopes in the S2 subunit. Examples include the FP (788–806 aa; 18 amino acids), HR2 (1127–1177 aa; 50 amino acids) and MPER (1204–1211 aa; 7 amino acids) [153]. These epitopes are too short to induce immune response except if they are used with adjuvants or fused with other immunogenic proteins as observed in the influenza HA stalk universal vaccine designs [102,104,154].

Despite the concerns mentioned above and other associated concerns, such as the antibody binding affinity, weak immunogenicity, etc., there are possible ways to increase the immunogenicity of the S2 protein.

Repeating epitopes or multiple epitopes: The fusion of the conserved epitopes can be used to develop an immunogen to induce broad and neutralizing antibodies. As observed in the licensed dengue multi-epitope-based vaccine Dengvaxia^®^, which carries the prM and E genes of different dengue virus serotypes, a high amount of B cells and T cells are induced [155]. We have also generated influenza vaccines that carried M2 ectodomains and/or HA stalk epitopes and could protect mice from influenza H1N1 and H3N2 challenges. In this study, four M2e sequences from human (two copies), swine (one copy) and bird (one copy) were fused by a GGG linker to form tM2e, while a copy of M2e from the human influenza strain was combined with the conserved regions of the HA stalk using a GSA linker to form HM2e. Then the tM2e or HM2e was further fused with an Ebola glycoprotein dendritic cell (DC)-targeting domain (E∆M) to form E∆M-tM2e or E∆M-HM2e, respectively. Animal studies showed that VSV carrying E∆M-tM2e or E∆M-HM2e mediated rapid and potent induction of M2 or/and HA antibodies in mice sera and mucosa [104]. This technique can also be used to fuse the selected conserved epitopes in the S1 region and/or S2 region into a multi-epitope-based vaccine.

Dendritic cell (DC)-targeting approach: The development of a vaccine using the DC-targeting approach has recently gained much attention [154,156,157]. Targeting SARS-CoV-2 S2 conserved antigens to antigen-presenting cells (APCs) could increase the immunogenicity of the antigens. In a study by Marlin et al., the SARS-CoV-2 RBD was targeted to the cluster of differentiation (CD)-40 (αCD40.RBD) to increase the immunogenicity of the RBD [158]. The formed vaccine candidate αCD40.RBD induced significantly high amounts of T and B cells in humanized mice, while a single dose of the αCD40.RBD rapidly increased broadly neutralizing antibodies in previously SARS-CoV-2-exposed convalescent non-human primates. Moreover, CoVs SPs can also be targeted to DCs by using CpG or CD205 (DEC-205) to increase the immunogenicity of the SPs [154,159]. Our recent publication has demonstrated another technology for targeting DCs with the use of the DC targeting domain of the Ebola GP [68,160,161]. We demonstrated that targeting the SARS-CoV-2 S2 subunit protein to DCs using the DC-targeting domain of the Ebola glycoprotein could induce protective immune responses in hamsters [68].

The use of adjuvants: Adjuvants, such as chemokine encoding plasmids, co-stimulatory molecule encoding plasmids, plasmids encoding pathogen-recognition receptor (PRR) ligands and immune-signalling molecules, can be incorporated into plasmids and either co-express with SARS-CoV-2 antigens or be administered separately during immunization [162]. Hui et al. demonstrated that the co-expression of IL-2 with the SARS-CoV S protein in a DNA vaccine increased the immunogenicity of the SP by inducing a higher amount of IgG than the SARS-CoV S protein alone [163]. In addition, their study revealed that the electroporation method of immunization had better immune responses than intramuscular or oral administration. Gary et al. also demonstrated that the co-formulation of the plasmid-encoded mucosal chemokine cutaneous T cell-attracting chemokine (pCTACK; CCL27) with the SARS-CoV-2 SP in a DNA vaccine increased the immunogenicity against SARS-CoV-2 and conferred 100% protection against the Delta VOC in mice [164]. An adjuvant can also act as a delivery system, e.g., the nanoparticles increase the immunogenicity of the conserved S2 epitopes to boost T cells’ and B cells’ immune responses [165,166,167]. The LNP-based vaccine can induce both Th1- and Th2-based immune responses with more dominant Th1-type B cells and biased Th2-type B cells [167,168]. Ma et al. developed a nanoparticle-based vaccine with the fusion of the self-assembling 24-mer ferritin with the RBD and HR or the RBD alone [147]. They showed that nanoparticle immunization in rhesus hACE2 transgenic mice reduced the lung viral loads and induced persistent neutralizing antibodies, T cells and B cells in rhesus macaques [166].

***The routes of administration:*** The effectiveness of vaccines also depends on the route of administration. Routes of vaccine administration include oral, intranasal, intramuscular and subcutaneous [169]. The route of vaccine delivery may affect the protection conferred by the vaccine. The current route of SARS-CoV-2 vaccine administration is intramuscularly, however, intranasal vaccination that targets the local mucosal site of infection would be preferable. Although intramuscular vaccination can induce systemic immune responses, the induced immune responses can only get to the low respiratory tracts and not the upper respiratory tracts, while intranasal immunization can induce immune responses to both the upper and lower part [170,171]. Moreover, the intranasal delivery of the SARS-CoV-2 S2 can induce memory T cells and IgA within the nasal mucosal, as well as systemic IgG antibodies [68,172]. Neutralizing antibodies produced at the site of infection can immediately protect against the further spread of the virus into the body. Hassan et al. [173] demonstrated that a single dose of intranasal immunization protected the upper and respiratory tracts against SARS-CoV-2. We have also demonstrated that intranasal administration could protect against influenza in mice using a VSV-based vaccine [68]. It is important to note that the type of vaccine can additionally influence the effectiveness of intranasal vaccine delivery. The intranasal administering of DNA vaccines and human-neutralizing antibodies protected the lungs but did not reduce the viral load [174]. Oral immunization can also induce mucosal immunization against SARS-CoV-2. We have shown that oral immunization could induce substantial immune responses against SARS-CoV-2 but not as strongly as intranasal immunization (not yet published). Another study has also demonstrated that the orally delivered *Saccharomyces cerevisae* SARS-CoV-2-based vaccine induced mucosal immune responses and T cells in mice [175]. More investigations are still required to validate the use of intranasal or oral administrations of the SARS-CoV-2 S2-based vaccine.

## 5. Conclusions

Herd immunity is achieved either through natural infection and/or immunization, and it is expected to reduce or effectively eliminate the infection in a community [176]. Despite the high threshold of COVID-19 infection and massive vaccination of the public, it still seems unlikely to achieve herd immunity due to the emergence of several variants of SARS-CoV-2 [177,178,179,180]. Developing a universal vaccine against all SARS-CoV-2 variants, other endemic CoVs and potential emerging CoVs looks promising with highly conserved epitopes on the SARS-CoV-2 SP, especially the S2 subunit which can induce broadly neutralizing antibodies and humoral and cell-mediated immune responses.

Although these conserved regions in the S2 can be weak immunogens or inaccessible, these antigens can be strengthened to increase their immunogenicity in vaccine design. Moreover, intranasal delivery of the SARS-CoV-2 S2 to the mucosal may be more reliable in inducing long-term immune responses. Although different in silico, immunoinformatic or bioinformatic and immunophenotyping studies have demonstrated the presence of conserved epitopes and their immunogenicity among CoVs, more in vivo studies are required to validate the broad immunogenicity of these epitopes for viral designs. Nevertheless, the SARS-CoV-2 S2 antigen can be used to develop a universal vaccine that could neutralize variants of SARS-CoV-2 as well as other betacoronaviruses. Though S2 is very promising, neutralizing epitopes on the S1 subunit is equally important. Therefore, in designing a multi-epitope universal antigen against SARS-CoV-2 variants, the highly conserved HR2 and other identified conserved epitopes on NTD, RBD, FP and HR1 should be considered.

## Figures and Tables

**Figure 1 vaccines-11-00545-f001:**
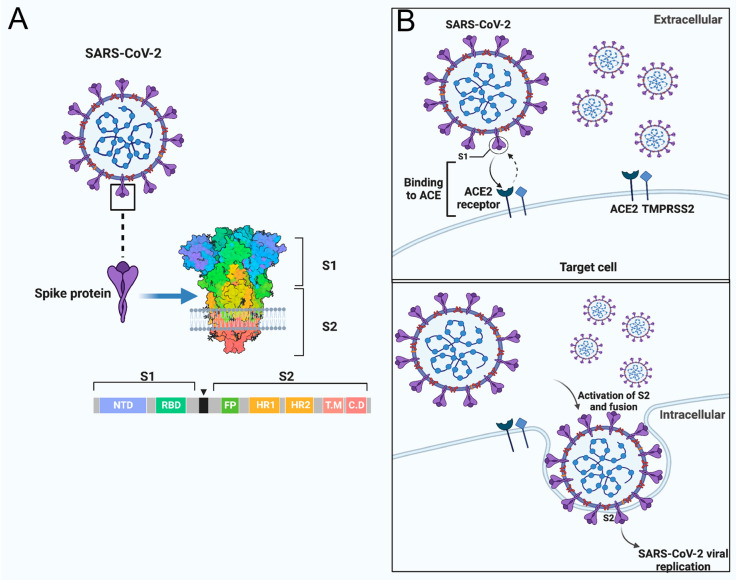
SARS-CoV-2 cell entry into the target cell. (**A**) SARS-CoV-2 virus with its SP. The SP contains the S1 and the S2 subunit. The S1 subunit comprises the NTD and RBD, while the S2 comprises the FP, HR1 and HR2. (**B**) The schematic summary of SARS-CoV-2 cell entry. SARS-CoV-2 binds to the host ACE receptor using the RBD domain of the S1 (upper panel). After the activation of the S2, SARS-CoV-2 uses the FP and HR for fusion and gains entry into the host (lower panel) [35,42]. Image created by Biorender.com. *Adapted from “SARS-CoV-2 Targeting of ACE2 Receptor and Entry in Infected Cell” and “An In-depth Look into the Structure of the SARS-CoV2 Spike Glycoprotein”, by BioRender.com (Accessed on 22 February 2023). Retrieved from https://app.biorender.com/biorender-templates*.

**Figure 2 vaccines-11-00545-f002:**
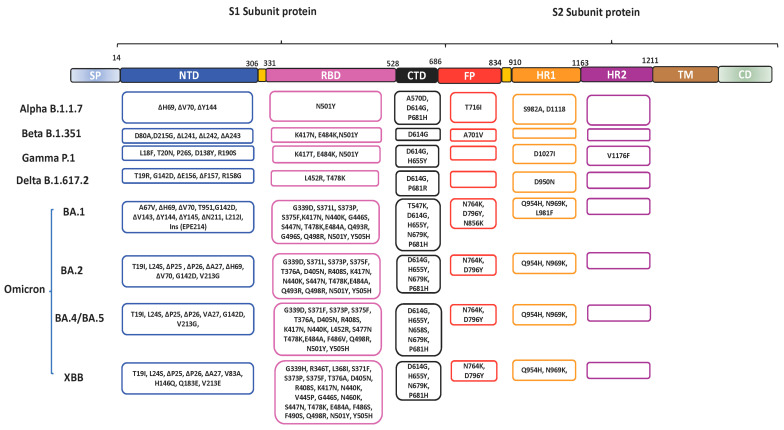
The schematic structure of the SARS-CoV-2 spike protein showing the mutations on the spike protein of selected SARS-CoV-2 variants of concern (VOC) including Alpha, Beta, Delta, Gamma and Omicron [6].

**Figure 3 vaccines-11-00545-f003:**
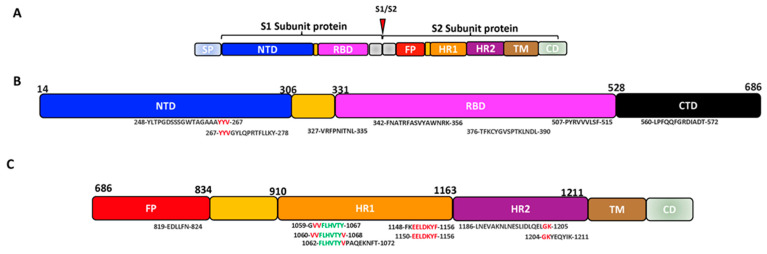
The schematic representation of some possible conserved epitopes on the SARS-CoV-2 SP (Black: positions occurring in a set of predicted epitopes; Red: positions occurring in 2 sets of predicted epitopes; Green: positions present in 3 sets of predicted epitopes). (**A**) Schematic structure of the SARS-CoV-2 spike protein. (**B**) Structure of the SARS-CoV-2 S1 subunit showing the conserved epitopes on the NTD and RBD. (**C**) Structure of the SARS-CoV-2 S2 subunit showing the conserved epitopes on the FP and HR.

**Table 1 vaccines-11-00545-t001:** Some predicted conserved epitopes on the SARS-CoV-2 S1 subunit that could induce neutralizing antibodies and/or adaptive immune responses.

Conserved Epitopes	Position	Immune Response Induced		Type of Study	Refs.
		B cells/Neutralizing antibodies	T cells		
YLTPGDSSSGWTAGAAAYYV	248–267 aa	Yes	Yes	Mathematically (in-house developed PERL scripts), in vivo	[116,120]
YYVGYLQPRTFLLKY	264–278 aa	NT	Yes	Web-based analytic tools	[119]
VRFPNITNL	327–335 aa	NT	Yes	Immunoinformatic, in vivo	[117,121]
FNATRFASVYAWNRK	342–356 aa	Yes	Yes	In silico, T-cell epitope mapping, molecular dynamics simulations, immunoinformatic	[118,122,123]
TFKCYGVSPTKLNDL	376–390 aa	Yes	Yes	Mathematically (in-house developed PERL scripts), bioinformatics, monoclonal antibody targeting	[116,124]
PYRVVVLSF	507–515 aa	NT	Yes	Immunoinformatic	[117]
LPFQQFGRDIADT	560–572 aa	Yes	Yes	PepSeq Analysis	[125]

NT—Not tested.

**Table 2 vaccines-11-00545-t002:** Some predicted conserved epitopes on the SARS-CoV-2 S2 subunit that could induce neutralizing antibodies and/or adaptive immune responses.

Conserved Epitopes	Position	Immune Response Induced		Type of Study	Refs.
		Neutralizing antibodies	T cells		
EDLLFN	819–824 aa	Yes	NT	Epitope-resolved profiling, structural and functional test	[125,126]
EELDKYF	1150–1156 aa	Yes	NT	Epitope-resolved profiling, structural and functional	[125,126]
GVVFLHVTY	1059–1067 aa	NT	Yes	Immunoinformatic	[117,127]
VVFLHVTYV	1060–1068 aa	NT	Yes	Immunoinformatic	[117,127]
FLHVTYVPAQEKNFT	1062–1072 aa	Yes	Yes	In silico, in vivo	[118]
SPDVDLGDISGINAS	1161–1175 aa	Yes	NT	In vivo	[128]
LNEVAKNLNESLIDLQELGK	1186–1205 aa	Yes	Yes	Mathematically (in-house developed PERL scripts), bioinformatic, in vivo	[116,120,129]
GKYEQYIK	1204–1211 aa	NT	NT	Antiviral inhibitory activity	[34]

NT—Not tested.

## Data Availability

Data sharing is not applicable.

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
