# Peer review of "Significance of Conserved Regions in Coronavirus Spike Protein for Developing a Novel Vaccine against SARS-CoV-2 Infection"

_vaccines, 2023, doi:10.3390/vaccines11030545_

Round 1

Reviewer 1 Report

Olukitibi et al. review the conserved zones of the SP from SARS-CoV-2 for universal vaccine development. The topic is interesting, but there is still work required for this review article to cover all the different topics needed to develop a SARS-CoV-2 vaccine using conserved regions of the spike protein.

Specific comments:

Line 31) Replace “effectivity” by “effectiveness”.

Figure 31-32) Authors should clarify is they consider the full pathogenic virus as a vaccine platform.

Line 41) What is “SP”?

Line 47) “an” or “and”?

Line 51-71) As a review, authors should include an image showing the infection process they describe for a better comprehension to the audience.

Line 106) Authors should also mention innate immune evasion by suppression of MHC-I and II class proteins as well as interfering with the interferon response.

Lien 114-115) CD4 T cells can also do this

Line 120-122) Are authors sure about this in regards to T-cell responses? Please check publications from Scripps.

Line 130-132) This is somewhat misleading. Authors should specify if most of these mutations happen in the RBD or in the S1 but outside of the RBD. This has different implications.

Line 133-170) Authors should significantly update this part (very outdated) since this has been already extensively described and information of new circulations must be included.

Line 159) What do authors mean by “resist”, prevention of breakthrough infection?

Line 214-215) Does this mean that different HLA alleles can recognize these epitopes? References should be included.

Table 2) Why does it say “neutralizing antibodies” instead of “yes” in the 1161-1175 aa epitope?

Line 258) Authors should clarify what a lipopeptide is.

Line 262-263) Do authors have references for this statement?

Section 3 – general comment) Authors should provide a table or even better, an image, showing the distinct conserved zones described in this section. As is, it is difficult to follow; an image would be greatly helpful for the audience.

Line 269-270) This depends more on the type of immunity induced for the vaccine (mucosal or systemic). Do authors have a reference for this, or is just speculation?

Line 280-284) Antibodies might be more cross-reactive against the S2, but I guess they have a lower neutralizing capacity in comparison to S1 antibodies. Authors should comment on this.

Line 298-303) The length of the epitope is not the sole factor, the accessibility, antibody affinity… also play role. Authors need to complement the information.

General comment) A section is missing regarding the routes of vaccine administration. This is highly important for the development of a universal vaccine since this will guide the type of immune response that follows.

Author Response

Response to Reviewer 1 Comments

Olukitibi et al. review the conserved zones of the SP from SARS-CoV-2 for universal vaccine development. The topic is interesting, but there is still work required for this review article to cover all the different topics needed to develop a SARS-CoV-2 vaccine using conserved regions of the spike protein.

Point 1: Line 31) Replace “effectivity” by “effectiveness”.

Response 1: Thank you for this Point. The word “effectivity” has now been changed to “effectiveness” (Line 32)

Point 2: Figure 31-32) Authors should clarify if they consider the full pathogenic virus as a vaccine platform.

Response 2: We thank the reviewer for this request. We are not considering the full pathogenic virus as a vaccine platform but the use of subunit proteins (antigenic epitopes) present on the pathogenic virus.

Point 3: Line 41) What is “SP”?

Response 3: Thank you for spotting this. SP stands for spike protein. This has been corrected as “the spike protein (SP)” (Line 41)

Point 4: Line 47) “an” or “and”?

Response 4: This is “an”. This is an article defining envelope protein

Point 5: Line 51-71) As a review, authors should include an image showing the infection process they describe for better comprehension to the audience.

Response 5: Thank you for this Point. The Figure has been included as Figure 1

Point 6: Line 106) Authors should also mention innate immune evasion by suppression of MHC-I and II class proteins as well as interfering with the interferon response.

Response 6: Thank you for your suggestion. We want our discussion to focus only on the SARS-CoV-2 SP. However, we have added some information in this regard to the manuscript (Lines 110-122)

“Noteworthily, mutations in the SARS-CoV-2 ORF8 (which could modulate the SP during transmission) have been associated with inhibition of the Major Histocompatibility Complex (MHC)- 1 stimulation thus, leading to the low production of cytotoxic T cells (CD8+ T cells)  [1-3]. Likewise, the SARS-CoV-2 SP has been demonstrated to regulate the innate immune response through the activation of the NF-κB pathway in human macrophages and monocytes, thus increasing the production of inflammatory responses (IL-8) [4]. In our study, the SARS-CoV-2 variant, Delta variant SP had the highest potential of stimulating the NF-κB pathway in macrophages to induce inflammatory cytokines in the monocyte-derived macrophages (MDMs) and Monocytes derived dendritic cells (MDDCs) when compared with the Wuhan strains or D614G variants. This suggests that continuous mutation of the SARS-CoV-2 SP also impacts the innate immune responses and could further increase virulence, pathogenicity, and immune evasion of the virus [5].”

Point 7: Line 114-115) CD4 T cells can also do this

Response 7: This statement has been revised to capture the function of both the CD4 T cells and the CD8 T cells. (Lines 129-133)

“The SARS-CoV-2 specific CD4+ T and CD8+ T cells help to trigger the induction of SARS-CoV-2 specific B cells that further generate most of the neutralizing antibodies as well as memory B cells and long-term humoral immunity. They also regulate antiviral activities, recruit innate cells, kill infected cells, and facilitate tissue repair [6-11].”

Point 8: Line 120-122) Are authors sure about this in regards to T-cell responses? Please check publications from Scripps.

Response 8: We thank the reviewer for this Point, we have revised adaptive immune responses to “the humoral immune responses and neutralizing antibodies” (Line 137- 138)

Point 9: Line 130-132) This is somewhat misleading. Authors should specify if most of these mutations happen in the RBD or in the S1 but outside of the RBD. This has different implications.

Response 9: Thank you for your Point. We meant to mention that most of the mutations are within the S1 protein. We have rephrased the sentence as “Most of the mutations are found in the S1 subunit protein”. (Line 147)

Point 10: Line 133-170) Authors should significantly update this part (very outdated) since this has been already extensively described and information on new circulations must be included.

Response 10: We thank the reviewer for this request. We have updated the information in this session as requested.

The following information has been added to the manuscript. We have also included a Figure summarizing the mutations in SARS-CoV-2 variants (Fig. 2).

“Most of the mutations are found in the S1 subunit protein. SARS-CoV-2 variants include the Alpha having N501Y mutation in RBD, Beta having N501Y, K417N, and E484K mutations in the RBD, Gamma with N501Y, K417T, and E484K, Delta having T478K, L452 in the RBD, and Omicron with S371L, G339D, S375F, S373P, K417N, N440K, S477N, G446S, E484A, T478K, Q493K, Q498R, G496S, N501Y, and Y505H in the RBD  (Figure 2) [12-14].” (Line 147-152)

“The T95I, N211I, V213R, Y505H, N786K, and N856K mutations in Omicron variants contribute to its pathogenicity and virulence [15]. Sub-variants of Omicron have also been identified as BA.1, BA.2, BA.2.12.1, BA.2.3, BA.2.9, BA.3, BA.4, BA.5 (BQ.1, BQ.1.1, BF.7, and BA.4.6) and recombinant BA.1/BA.2 [12, 16]. The BA.2 progressively replaced BA.1 which was the most prevalent globally while BA.3 spread is restricted. The BA.4 and BA.5 which were first isolated in South Africa replaced the BA.2 variant having mutations F486V and R493Q and deletion in positions 69 and 70 in the RBD [17, 18].” (Line 200-207).

“The Omicron subvariants shared many mutations among themselves including the P681H, H655Y, N679K, and D614G mutations in the S1 subunit; G339D, S371L/F, S373P, S375F, K417N, N440K, S477N, T478K, E484A, Q498R, N501Y and Y505H, in RBD; G142D in N-terminal domain (NTD) and N969K, N764K, D796Y and Q954H in the S2 subunit (Figure 2) [19]. Omicron BA.1 and BA.2 have 12 common mutations on the RBD which are G339D, S373P, S375F, K417N, N440K, S477N, T478K, E484A, Q498R, Q493R, N501Y and Y505H whereas BA.2 has a total of 31 mutations in the SP including the unique R408S, and S371F (Figure 2). However, the BA.3 also has unique mutations T376A and D405N in the RBD [18].” (Line 215-223)

“Importantly, the Omicron BA.1, BA.2 and BA.3 are very good at escaping immune responses induced by natural infection or by one or two doses of the vaccine while a third dose can give partial protection [12]. Although the SARS-CoV-2 bivalent vaccine (wildtype and Omicron) can provide long-term protection, a BA.1 derived vaccine may not provide broad protection against SARS-CoV-2 variants [20]. Furthermore, the Omicron BA.4, BA.5 and BA.2.12.1 have been demonstrated to show more immune response escape strategies than the BA.1 and BA.2 due to the presence of F486V and D405N mutations [20, 21]. Cao et al. [20] demonstrated that structurally, the SP of BA.4, BA.5 and BA.2.12.1 have the same binding affinity with the ACE-2. They further revealed that BA.4, BA.5 and BA.2.12.1 could evade neutralizing antibodies from the plasma of BA.1 infected immunized patients or triple-vaccinated individuals more than BA.2.” (Line 250-262)

Point 11: Line 159) What do authors mean by “resist”, prevention of breakthrough infection?

Response 11: We have rephrased the statement as: “Therefore, it is not surprising to find that other variants of SARS-CoV-2 such as Beta (B.1.351), Gamma (B.1.1.28), Delta (B.1.617.2), and especially Omicron (B.1.1.529) to substantially escape the immune responses induced by the licensed vaccines” (Line 236-238)

Point 12: Line 214-215) Does this mean that different HLA alleles can recognize these epitopes? References should be included.

Response 12: thank you for the observation. We have revised this as “most HLA alleles” and the reference has been included as PMID: 35171086 (doi: 10.1080/22221751.2022.2043727)

Point 13: Table 2) Why does it say “neutralizing antibodies” instead of “yes” in the 1161-1175 aa epitope?

Response 13: Thank you for spotting that. We have corrected it to “yes”

Point 14: Line 258) Authors should clarify what a lipopeptide is.

Response 14: A lipopeptide in this context is a lipid that contains the MPER peptide. We have revised this statement as “Yu et al. have demonstrated that cholesterol-conjugated lipopeptide containing SARS-CoV-2 MPER prepared by using cholesteryl succinate monoester could inhibit viral entry indicating the importance of MPER in viral entry and fusion” (Line 355-358)

Point 15: Line 262-263) Do authors have references for this statement?

Response 15: Reference for this statement has been included

Point 16: Section 3 – general Point) Authors should provide a table or even better, an image, showing the distinct conserved zones described in this section. As is, it is difficult to follow; an image would be greatly helpful for the audience.

Response 16: We thank the author for this Point. We have included a Figure showing the various conserved epitopes on the SARS-CoV-2 as Figure 3.

Point 17: Line 269-270) This depends more on the type of immunity induced for the vaccine (mucosal or systemic). Do authors have a reference for this, or is just speculation?

Response 17: Thank you for your Point. This was just speculation and that is why we use the word “might”. The following statement justifies our speculation.

Point 18: Line 280-284) Antibodies might be more cross-reactive against the S2, but I guess they have a lower neutralizing capacity in comparison to S1 antibodies. Authors should Point on this.

Response 18: We thank the reviewer for this question. The S1 could induce stronger neutralizing antibodies but might not be broad due to mutation, However, the S2 neutralizing epitopes can induce broader neutralizing antibodies but may not be as strong as the S1 epitopes. We have included this information as follows

“Notably, S1 could induce stronger neutralizing antibodies but might not be broad due to mutation, However, the S2 neutralizing epitopes can induce broader neutralizing antibodies but may not be as strong as the S1 epitopes” (Line 376-380)

Point 19: Line 298-303) The length of the epitope is not the sole factor, the accessibility, antibody affinity… also play role. Authors need to complement the information.

Response 19: We thank the reviewer for this Point. However, we have already discussed the inaccessibility as a challenge under “structural positioning” (Line 345-350). We agreed that there could be other possible challenges associated with the use of the S2 antigen for vaccine development that is not covered in this review. We have therefore revised this statement “Despite the concerns mentioned above and other associated concerns such as the antibody binding affinity, weak immunogenicity etc., there are possible ways to increase the immunogenicity of the S2 protein.” (Line 409-411)

Point 20: General Point) A section is missing regarding the routes of vaccine administration. This is highly important for the development of a universal vaccine since this will guide the type of immune response that follows.

Response 20: We thank the reviewer for this Point. This section has been included in the main manuscript.

The routes of administration: The effectiveness of vaccines also depends on the route of administration. “Routes of vaccine administration include oral, intranasal, intramuscular and subcutaneous  [22]. The route of vaccine delivery may affect the protection conferred by the vaccine. The current route of SARS-CoV-2 vaccine administration is intramuscularly, however, intranasal vaccination which targets the local mucosal site of infection would be preferable. Although intramuscular vaccination can induce systemic immune responses, the induced immune responses can only get to the low respiratory tracts and not the upper respiratory tracts. While intranasal immunization can induce immune responses to both the upper and lower part. [23, 24]. Moreover, the intranasal delivery of SARS-CoV-2 S2 can induce memory T cell, and IgA within the nasal mucosal as well as systemic IgG antibodies [25, 26]. Neutralizing antibodies produced at the site of infection can immediately protect against the further spread of the virus into the body. Hassan et al. [27] demonstrated that a single dose of intranasal immunization protected the upper and respiratory tracts against SARS-CoV-2. We have also demonstrated that intranasal administration could also protect against influenza in mice using a VSV-based vaccine [26]. It is important to note that type of vaccine can as well influence the effectiveness of intranasal vaccine delivery. The intranasal administering of DNA vaccine and human-neutralizing antibodies protected the lungs but did not reduce the viral load [28]. Oral immunization can as well induce mucosal immunization against SARS-CoV-2. We have shown that oral immunization could induce substantial immune responses against SARS-CoV-2 but not as strong as intranasal immunization (Not yet published). Another study has also demonstrated that the orally delivered Saccharomyces cerevisae SARS-CoV-2 based vaccine-induced mucosal immune responses and T cells in mice  [29]. More investigations are still required to validate the use of intranasal or oral administrations of SARS-CoV-2 S2 based vaccine.” (Line 463-487ß) 

References

  1. Chou, J.-M., et al., The ORF8 protein of SARS-CoV-2 modulates the spike protein and its implications in viral transmission. Frontiers in microbiology, 2022. 13.
  2. Rubio-Casillas, A., E.M. Redwan, and V.N. Uversky, SARS-CoV-2: a master of immune evasion. Biomedicines, 2022. 10(6): p. 1339.
  3. Moriyama, M., et al., SARS-CoV-2 Omicron subvariants evolved to promote further escape from MHC-I recognition. bioRxiv, 2022: p. 2022.05. 04.490614.
  4. Dosch, S.F., S.D. Mahajan, and A.R. Collins, SARS coronavirus spike protein-induced innate immune response occurs via activation of the NF-κB pathway in human monocyte macrophages in vitro. Virus research, 2009. 142(1-2): p. 19-27.
  5. Taefehshokr, N., et al., Covid-19: perspectives on innate immune evasion. Frontiers in immunology, 2020. 11: p. 580641.
  6. Liu, X., R.I. Nurieva, and C. Dong, Transcriptional regulation of follicular T‐helper (Tfh) cells. Immunological reviews, 2013. 252(1): p. 139-145.
  7. Kaneko, N., et al., Loss of Bcl-6-expressing T follicular helper cells and germinal centers in COVID-19. Cell, 2020. 183(1): p. 143-157. e13.
  8. Lu, X., et al., Identification of conserved SARS-CoV-2 spike epitopes that expand public cTfh clonotypes in mild COVID-19 patients. Journal of Experimental Medicine, 2021. 218(12).
  9. Crotty, S., T follicular helper cell differentiation, function, and roles in disease. Immunity, 2014. 41(4): p. 529-542.
  10. Sette, A. and S. Crotty, Adaptive immunity to SARS-CoV-2 and COVID-19. Cell, 2021. 184(4): p. 861-880.
  11. Crotty, S., T follicular helper cell biology: a decade of discovery and diseases. Immunity, 2019. 50(5): p. 1132-1148.
  12. Chatterjee, S., et al., A Detailed Overview of SARS-CoV-2 Omicron: Its Sub-Variants, Mutations and Pathophysiology, Clinical Characteristics, Immunological Landscape, Immune Escape, and Therapies. Viruses, 2023. 15(1): p. 167.
  13. Cherian, S., et al., Convergent evolution of SARS-CoV-2 spike mutations, L452R, E484Q and P681R, in the second wave of COVID-19 in Maharashtra, India. Microorganism, 2021. 9(7): p. 1542-1553.
  14. Cherian, S., et al., SARS-CoV-2 spike mutations, L452R, T478K, E484Q and P681R, in the second wave of COVID-19 in Maharashtra, India. Microorganisms, 2021. 9(7): p. 1542.
  15. Kumar, S., K. Karuppanan, and G. Subramaniam, Omicron (BA. 1) and sub‐variants (BA. 1.1, BA. 2, and BA. 3) of SARS‐CoV‐2 spike infectivity and pathogenicity: A comparative sequence and structural‐based computational assessment. Journal of Medical Virology, 2022. 94(10): p. 4780-4791.
  16. Mohapatra, R.K., et al., The recently emerged BA. 4 and BA. 5 lineages of Omicron and their global health concerns amid the ongoing wave of COVID-19 pandemic–Correspondence. International Journal of Surgery (London, England), 2022. 103: p. 106698.
  17. Callaway, E., Are COVID surges becoming more predictable. Nature, 2022. 605.
  18. Tegally, H., et al., Continued emergence and evolution of Omicron in South Africa: New BA. 4 and BA. 5 lineages. MedRxiv, 2022: p. 2022.05. 01.22274406.
  19. Mohseni Afshar, Z., et al., SARS-CoV-2 Omicron (B. 1.1. 529) Variant: A Challenge with COVID-19. Diagnostics, 2023. 13(3): p. 559.
  20. Cao, Y., et al., BA. 2.12. 1, BA. 4 and BA. 5 escape antibodies elicited by Omicron infection. Nature, 2022. 608(7923): p. 593-602.
  21. Cao, Y., et al., Omicron escapes the majority of existing SARS-CoV-2 neutralizing antibodies. Nature, 2022. 602(7898): p. 657-663.
  22. Clark, E.M. and M.M. Pippin, Safe And Effective Administration Of Vaccines And Epinephrine Autoinjection, in StatPearls [Internet]. 2022, StatPearls Publishing.
  23. Park, J.H. and H.K. Lee, Delivery routes for COVID-19 vaccines. Vaccines, 2021. 9(5): p. 524.
  24. van Doremalen, N., et al., Intranasal ChAdOx1 nCoV-19/AZD1222 vaccination reduces shedding of SARS-CoV-2 D614G in rhesus macaques. BioRxiv, 2021.
  25. Gebhardt, T., et al., Memory T cells in nonlymphoid tissue that provide enhanced local immunity during infection with herpes simplex virus. Nature immunology, 2009. 10(5): p. 524-530.
  26. Ao, Z., et al., A Recombinant VSV-Based Bivalent Vaccine Effectively Protects against Both SARS-CoV-2 and Influenza A Virus Infection. Journal of Virology, 2022: p. e01337-22.
  27. Hassan, A.O., et al., A single-dose intranasal ChAd vaccine protects upper and lower respiratory tracts against SARS-CoV-2. Cell, 2020. 183(1): p. 169-184. e13.
  28. Zhou, D., et al., Robust SARS-CoV-2 infection in nasal turbinates after treatment with systemic neutralizing antibodies. Cell host & microbe, 2021. 29(4): p. 551-563. e5.
  29. Gao, T., et al., Immune response induced by oral administration with a Saccharomyces cerevisiae-based SARS-CoV-2 vaccine in mice. Microbial Cell Factories, 2021. 20(1): p. 95.

Reviewer 2 Report

Olukitibi et al. described the role of conserved regions in the coronavirus spike protein, which is utilized to develop novel vaccines against, especially SARS-CoV-2 infection. This review article covers almost everything but lacks structural details. Many mutations were described in this review. For easy understanding of the readers, authors should include the relevant 3D structures (all are available in the protein databank) and map the observed mutations in the appropriate subunits of SP. This manuscript may be accepted for publication after the incorporation of structural diagrams.

Author Response

Response to Reviewer 2 Comments

Olukitibi et al. described the role of conserved regions in the coronavirus spike protein, which is utilized to develop novel vaccines against, especially SARS-CoV-2 infection. This review article covers almost everything but lacks structural details. Many mutations were described in this review.

Point 1: For easy understanding of the readers, authors should include the relevant 3D structures (all are available in the protein databank) and map the observed mutations in the appropriate subunits of SP. This manuscript may be accepted for publication after the incorporation of structural diagrams.

Response 1: We thank the author for their comments. We have included a figure (Figure 2) showing different mutations on the SP of SARS-CoV-2 variants. We apologize that we cannot put this Figure in the 3D format as we don’t have the expertise in our lab.

Reviewer 3 Report

The authors aimed, in a review, to shed light on the conserved epitopes capable of inducing a broad immune response against multiple CoVs and highlight how to circumvent the associated possible challenges with lessons gleaned from SARS-CoV-2.

The study covers some issues that have been overlooked in other similar topics. The structure of the manuscript appears adequate and well divided in the sections. Moreover, the study is easy to follow, but some issues should be improved. Some of the comments that would improve the overall quality of the study are:

I-) Authors must pay attention to the technical terms acronyms they used in the text

II-) Conclusion Section: This paragraph required a general revision to eliminate redundant sentences and to add some "take-home message".

Author Response

Response to Reviewer 3 Comments

The authors aimed, in a review, to shed light on the conserved epitopes capable of inducing a broad immune response against multiple CoVs and highlight how to circumvent the associated possible challenges with lessons gleaned from SARS-CoV-2.

The study covers some issues that have been overlooked in other similar topics. The structure of the manuscript appears adequate and well divided in the sections. Moreover, the study is easy to follow, but some issues should be improved. Some of the comments that would improve the overall quality of the study are:

Point 1: I-) Authors must pay attention to the technical terms acronyms they used in the text

Response 1: We thank the reviewer for their comments. We have defined all the acronyms used in the manuscript

Point 2: II-) Conclusion Section: This paragraph required a general revision to eliminate redundant sentences and to add some "take-home message".

Response 2: Thank you for your comments. We have revised the conclusion to remove redundant statements and have added a take-home message. (Line 490-509).

“Herd immunity is achieved either through natural infection and/or immunization and it is expected to reduce or effectively eliminate the infection in a community [1]. Despite the high threshold of COVID-19 infection and massive vaccination of the public, it still seems unlikely to achieve herd immunity due to the emergence of several variants of SARS-CoV-2 [2-5]. Developing a universal vaccine against all SARS-CoV-2 variants, other endemic CoVs and potential emerging CoVs looks promising with highly conserved epitopes on the SARS-CoV-2 SP, especially the S2 subunit which can induce broadly neutralizing antibody, humoral and cell-mediated immune responses.

However, these conserved regions in S2 can be weak immunogens or inaccessible, these antigens can be strengthened to increase their immunogenicity in vaccine design. Moreover, intranasal delivery of the SARS-CoV-2 S2 to the mucosal may be more reliable in inducing long-term immune responses. Although different in silico, immunoinformatic or bioinformatic, and immunophenotyping studies have demonstrated the presence of conserved epitopes immunogenicity among CoVs, more in vivo studies are required to validate the broad immunogenicity of these epitopes for viral designs. Nevertheless, the SARS-CoV-2 S2 antigen can be used to develop a universal vaccine that could neutralize variants of SARS-CoV-2 as well as other betacoronaviruses. Though S2 is very promising, neutralizing epitopes on the S1 subunit are equally important. Therefore, in designing a multi-epitope universal antigen against SARS-CoV-2 variants, the highly conserved HR2 and other identified conserved epitopes on NTD, RBD, FP and HR1 should be considered”

References

  1. Desai, A.N. and M.S. Majumder, What is herd immunity? JAMA, 2020. 324(20): p. 2113-2113.
  2. Aschwanden, C., Five reasons why COVID herd immunity is probably impossible. Nature, 2021: p. 520-522.
  3. Morens, D.M., G.K. Folkers, and A.S. Fauci, The concept of classical herd immunity may not apply to COVID-19. The Journal of Infectious Diseases, 2022.
  4. Ashton, J., COVID-19 and herd immunity. Journal of the Royal Society of Medicine, 2022. 115(2): p. 76-77.
  5. MacIntyre, C.R., V. Costantino, and M. Trent, Modelling of COVID-19 vaccination strategies and herd immunity, in scenarios of limited and full vaccine supply in NSW, Australia. Vaccine, 2022. 40(17): p. 2506-2513.